# Composite Aramid Membranes with High Strength and pH-Response

**DOI:** 10.3390/polym13040621

**Published:** 2021-02-19

**Authors:** Xiao Wang, Shi Li, Yuanyuan Tu, Jiwen Hu, Zhenzhu Huang, Shudong Lin, Xuefeng Gui

**Affiliations:** 1Guangzhou Institute of Chemistry, Chinese Academy of Sciences, Guangzhou 510650, China; wangxiao3291@163.com (X.W.); lichunsi002@163.com (S.L.); tyy0301@163.com (Y.T.); kas1230@126.com (Z.H.); linshudong71@163.com (S.L.); guixf@gic.ac.cn (X.G.); 2University of Chinese Academy of Sciences, Beijing 100049, China; 3CAS Engineering Laboratory for Special Fine Chemicals, Guangzhou 510650, China; 4Incubator of Nanxiong CAS Co., Ltd., Nanxiong 512400, China; 5Guangdong Provincial Key Laboratory of Organic Polymer Materials for Electronics, Guangzhou 510650, China

**Keywords:** aramid nanofiber, hydrolyzed aramid nanofiber, vacuum filtration, dual pH-responsive characteristic, high strength

## Abstract

The pH-responsive membrane is a new wastewater treatment technology developed in recent years. In this paper, a novel film with intelligent pH-responsiveness was first prepared by blending functional gates comprised of hydrolyzed aramid nanofibers (HANFs) into aramid nanofiber (ANF) membranes via a vacuum filtration method. Those as-prepared membranes exhibited dual pH-responsive characteristics, which were featured with a negative pH-responsiveness in an acidic environment and a positive pH-responsiveness in basic media. These dual pH-responsive membranes also exhibited a high tensile strength which could still reach 55.74 MPa (even when the HANFs content was as high as 50 wt%), a high decomposition temperature at ~363 °C, and good solvent resistance. The membranes described herein may be promising candidates for a myriad of applications, such as the controlled release of drugs, sensors, sewage treatment, etc.

## 1. Introduction

As one of the environmental responsiveness of “smart” membranes, pH-responsive membranes can adjust their flux and separation performance with the change of environmental pH value [1,2,3]. These characteristics allow the pH-responsive membranes to be used in many fields, especially water and wastewater treatment and desalination, drug release biotechnology and food industry [4,5,6].

As described in recent reports, pH-responsive membranes can be fabricated by various methods, such as plasma irradiation, high-energy radiation, ultraviolet irradiation, surface grafting, and physical blending [7,8,9]. Li et al. introduced polyacrylic acid to the surface of nylon films via atom-transfer radical polymerization to get a pH-responsive membrane that exhibited pH-responsiveness over a pH range from 3 to 8 [10]. Yi et al. prepared positive pH-responsive polyethersulfone porous films by extending poly (N-methacrylic acid) N-N-dimethylaminoethyl ester block to F127 side segment and then applying it to modified polyethersulfone porous films [11]. Mondal et al. synthesized the pH responsive poly-(vinylidene-fluoride-co-hexafluoro-propylene) membrane and phase inversion technique for a permeation study of glucose in the presence of various salts [2]. However, the surface grafting often leads to balance between the flux and reaction performance, and this method is time-consuming and complex. In contrast, physical mixing before membrane formation is efficient and productive [12,13].

As we all know, aramid fibers have been widely used in advanced composite materials, due to its high strength, high modulus and excellent heat and aolvent resistance [14,15,16,17,18,19]. Additionally, more and more attention has been paid to the study of aramid nanofiber (ANF) as an additive to strengthen other membranes or as a base membrane to prepare a functional membrane [20]. Bruggen et al. prepared an m-phenylenediamine-based thin-film composite nanofiltration membrane with excellent desalination performance by interfacial polymerization on a solvent-resistant Kevlar nanofibrous hydrogel substrate [21]. Surprisingly, there have been no reports about pH-responsive membranes prepared by Kevlar fibers. However, on account of the lack of polar functional groups, high crystallinity, and the tendency of hydrogen bonds to form between molecules, aramid fibers are chemically inert and their surfaces are extremely smooth. Consequently, their interfacial bonding strength with other substrates is poor, which greatly limits the applicability of aramid fibers [22,23,24].

As described herein, we have prepared a novel aramid nanofiber film with dual pH-responsive behavior that was comprised of aramid nanofibers (ANFs) and hydrolyzed aramid nanofibers (HANFs). The pH-responsive membranes prepared in this study can not only overcome the poor binding energy between ANF and matrix interfaces, but also retain much of the excellent performance of ANFs. Interestingly, we found that the obtained films have excellent distinctive properties such as smart dual pH-responsive characteristics, desirable mechanical behavior, as well as excellent physical and chemical stability. So far, we have not found any relevant reports.

## 2. Experimental

### 2.1. Materials

Aramid (Kevlar@49) threads were purchase from Dupont™, Wilmington, DE, US. Prior to use, the aramid fibers were cut into chopped fibers with lengths of ~5 mm, and washed with ultrasonication in acetone and ethanol for 3 h in sequence. After ultrasonication, chopped fibers were washed with water and dried in an oven at 80 °C. Dimethyl sulfoxide (DMSO, 99%) was obtained from Sigma-Aldrich, Shanghai, China. Methanol (99.5%), hexane (95%), sodium hydroxide (NaOH, 99% ), sodium chloride (NaCl, 99.5%), potassium *tert*-butoxide (KTB, 98%), N-methyl pyrrolidone (NMP, 99.5%), dichloromethane (DMC, 99.5%), N,N-dimethylformamide (DMF, 98% ), dimethylacetamide (DMAc, 98%), and bovine serum albumin (BSA, 98%) were provided by Aladdin, shanghai, China. Hydrochloric acid (HCl, 36~38%), sulfuric acid (H_2_SO_4_, 95~98%), nitric acid (HNO_3_, 65~68%), ethanol (99.7%), acetone (99.7%), toluene (99.5%), tetrahydrofuran (THF, 99.5%) were obtained from Reagent Co. LTD, Guangzhou, China. The distilled water was prepared in our laboratory. All of the above reagents were used without any further purification.

### 2.2. Preparation of Membrane

#### 2.2.1. Preparation of the Solution of ANF

The preparation of the ANF dispersion (0.2 g/100 mL) was referred to the method that had been previously developed by our research group [25]. A total of 0.2152 g of methanol and 0.2488g of KTB (the amount of methanol and KTB was 4 times that of amide bond in aramid threads, respectively) were added to a 250 mL single-port flask, and then 100.0 mL of DMSO were add to that flask. After magnetic stirring at 70 °C for 3 h, a transparent dispersion was obtained and then it was cooled to 25 °C. Subsequently, 0.2000 g of treated Kevlar chopped fibers were added into that dispersion solution. Finally, a uniform dark red dispersion (ANF/DMSO) was obtained via magnetic stirring at 25 °C for 48 h.

#### 2.2.2. Preparation of the ANF Hydrogel

Firstly, the ANF hydrogel was precipitated from solution by adding a proton donor distilled for proton recovery [26]. Distilled water (100.0 mL) was added to a 100.0 mL of ANF/DMSO dispersion bit-by-bit (the dripping rate was 3 to 4 drops per second), and continuously stirred for 2 h to obtain a colloid-like orange system. Finally, the colloid-like orange system was washed with distilled water, the volume of which was equivalent to five times that of that colloid-like orange system by means of filtration, until the KTB, methanol, and DMSO were thoroughly removed from the ANF system and the pH of the above system returned to neutral. The ANF hydrogels (the content of ANF was 3.75 wt%) was finally obtained.

#### 2.2.3. Preparation of the HANF Hydrogel

The HANFs were prepared via the method described by Lili Lv and coworkers [15]. The ANF hydrogels (the content of ANF was 0.2000 g) were added into a mixture acid of HNO_3_ and H_2_SO_4_. The total mass of the above system was 100.0000 g and the mass fraction of HNO_3_ and H_2_SO_4_ was 8.75 wt% and 37.5 wt%, respectively. After that, the above system was magnetically stirred at 90 °C for 1 h. Subsequently, it was rapidly cooled to 25 °C by a low-constant temperature tank (DC-0506), and washed with distilled water, the volume of which was equivalent to five times that of above system to remove the residual H_2_SO_4_ and HNO_3_, until the pH returned to neutral. The HANF hydrogel (the content of HANF was 3.75 wt%) was obtained and the yield of HANF was 83.3%.

#### 2.2.4. Preparation of the ANF/HANF Composite Membranes

The ANF/HANF composite membranes were manufactured by means of vacuum suction filtration. Firstly, ANF/HANF membranes (HANFs: 10 wt% of ANF) were prepared. In total, 0.0533 g of HANF hydrogel (the weight of HANFs was about 0.0020 g) were added to 10.0 mL distilled water, the volume of which was the same as that of the ANF/DMSO dispersion. After thorough mixing, the HANFs suspension was added to 10 mL ANF/DMSO dispersion bit-by-bit (the dripping rate was 3 to 4 drops per second) in order to separate the ANF, and stirring was continued for 12 h to ensure that a uniform colloid-like state was obtained. Subsequently, the membranes were fabricated by vacuum-filtrating the above mixed dispersion system with a polyvinylidene fluoride microfiltration membrane (pore size: 0.22 μm, diameter: 50 mm) for 30 min. Then the membrane was washed with about 50.0 mL of deionized water by vacuum filtration until DMSO and other small molecule substances remaining in the film were removed. The film was then released from the microfiltration membrane with 10.0 mL of acetone. After that, the membrane was completely immersed in 30.0 mL of ethanol for 12 h to remove water and acetone remaining in the membranes, and then was immersed in 30.0 mL of *n*-hexane for 12 h to remove ethanol remaining in the membranes. Finally, the films were fixed with filter paper, dried in a vacuum oven at 40 °C 12 h, and stored under dry conditions. Additionally, other ANF/HANF membranes (HANFs: 0, 20, 30, 40, and 50 wt% of ANF wt%) were prepared by the above method.

### 2.3. Characterization

#### 2.3.1. Fourier-Transform Infrared (FTIR) Spectroscopy

The FTIR spectra of the membranes were recorded with a Thermo Is50 FTIR spectrophotometer (Waltham, MA, US) and the range of the wavenumber was from 4000 to 400 cm^−1^.

#### 2.3.2. Atomic Force Microscope (AFM) Observation

AFM observation of the ANF, ANF hydrogels, HANFs, and membrane samples were performed as follows. The ANF and HANF dispersions were diluted to a concentration less than 0.01 wt% using DMSO and H_2_O, respectively. Then diluted dispersions were adhered to the surface of dried silicons wafers by spraying. The silicons wafers were ultrasonically cleaned with acetone, ethanol for 30 min successively and blow-dry with a nitrogen flow prior to use. Subsequently, the silicon wafer coated with the ANF was soaked into distilled water for 1 min and blow-dry with nitrogen. The silicon wafers coated with ANF hydrogels and HANF samples were blow-dried with nitrogen directly. The films samples were adhered to this sample stage directly. Then the samples were observed by a MultiMode 8 SPM (Bruker, karlsruhe, Germany) AFM instrument in tapping mode.

#### 2.3.3. Scanning Electron Microscope (SEM) Observation

The morphological qualities of the ANF and ANF/HANF films were surveyed using an S-4800 field emission scanning electron microscope (FESEM, Hitachi, Tokyo, Japan) with an accelerating voltage at 2.0 kV. The samples were prepared by rapidly quenching the membrane in liquid nitrogen.

#### 2.3.4. Water Flux at Different pH Values

The water flux of membranes was performed using a Millipore 8010 stirred ultra-filtration cell (Millipore Corporation, Bedford, MA, USA) under the pressure of 0.1 MPa at room temperature. The range of pH values was varied from 2.0 to 13.0 via adjustment with HCl (0.1 M) and NaOH (0.1 M). The ion concentrations of the above solutions were adjusted with a 0.005 M NaCl solution. Firstly, the membranes were cut into a circular shape with a diameter of 12 mm. Then the circular sample was fixed to the bottom of the Millipore 8010 stirred ultra-filtration cell. Prior to each test, the membrane was washed with aqueous solution with different pH values at least 30 min at an operating pressure of 0.2 MPa until the water flux was completely constant, and then the operating pressure was reduced to 0.1MPa. The mass of the aqueous solution with different pH values was recorded at 30 min intervals. At least five water flux measurements were taken for each membrane at different pH value in order to obtain an average and different value. In order to ensure the reliability of the numerical value, we carried out the same test on five membranes.

#### 2.3.5. BSA Retention Tests of Membranes at Different pH Values

Firstly, BSA solutions with different concentrations (50, 100, 150, 200, 250, 300, 350, 400, 450, 500, 600, 700, 800, 900, and 1000 mg/L) and different pH values were prepared, and then the absorbance of these BSA solutions were recorded at 280 nm by the 1800 ultraviolet visible spectrometer (Shimadzu, Japan). Standardization curves of the relationship between the absorbance and concentration of BSA were then obtained.

The BSA retention ratio was also measured in a Millipore 8010 ultrafiltration cell with stirring. The BSA solution (1000 mg/L) with different pH values was used in this test, and the operation pressure was 0.1 MPa. The diameter of the effective membrane area for this test was 12 mm. The typical operated process was similar to that used to determine the water flux. Firstly, a membrane was pre-pressed under 0.2 MPa with deionized water for at least 30min to obtained a stable flux, and then the pressure was reduced to 0.1 MPa, and the BSA solution (1000 mg/L) at a predetermined pH value was used to replace the deionized water. Before each measurement, the membrane was operated for at least 30 min. Then 5 mL filtrate was collected and the absorbance at 280 nm was recorded using a UV-vis spectrometer. The BSA retention ratio (*R*, %) was obtained by Equation (1):(1)$$R=\left(1-\frac{C_{p}}{C_{0}}\right)\times 100\%$$
where *C*_p_ and *C*_0_ were the BSA concentration in filtrate and feed solution, respectively. Each test at different pH values underwent five trials. After each trial, the membrane was washed using ethanol and deionized water, respectively. In order to ensure the reliability of the numerical value, we also carried out the same test on five membranes.

#### 2.3.6. Thermogravimetric Analysis (TGA)

The TGA of ANF, HANFs, and ANF/HANF membranes were carried on a TG209 TGA analyzer (Netzsch, Bavarian Asia, Germany) under a nitrogen atmosphere with a flow rate of 50 mL/min. The test temperature range was from 40 to 800 °C and the heating rate was 10 °C/min.

#### 2.3.7. Tensile Strength Tests

The tensile strength tests were performed on a CMT 7503 Universal Testing Machine (Shenzhen SANS Testing Machine Co., Ltd., Shenzhen, China) at a tensile rate of 20 mm/min. The samples were cut into rectangles with a length of 40 mm and a width of 5 mm. Each specimen subsequently underwent five trials to reduce error.

The tensile stress (σ) was calculated by Equation (2):(2)σ=F/S
where *F* was the tensile force and *S* was the cross-section area.

The tensile strain was calculated by *L*/*L*_0_, where *L* and *L*_0_ are the stretching distance and the original length.

#### 2.3.8. Chemical Resistance Tests

The chemical resistance of specimens was determined by immersing dry membranes which were accurately measured and weighted into various solvents and kept for 15 days. These solvents with different polarity included DMSO, NMP, DMC, THF, DMF, DMAc, toluene, methanol, and deionized water. After soaking, the residue was wiped off the samples with filter paper. The changes in the sizes of samples was measured. Additionally, the weight of membrane was measured after it had been dried under vacuum. After that, the water flux and BSA retention ratio of the dried membrane were tested referring the method of Section 2.3.4 and Section 2.3.5.

#### 2.3.9. Transparency Measurements

The transmission spectra of the membranes were carried on an optical transmittance meter (LS108H Shenzhen Linshang Technology Co., Ltd., Shenzhen, China). The measured wavelength was from 400 to 800 nm.

## 3. Results and Discussion

### 3.1. Preparation of the ANFs, HANFs, and ANF/HANF Membranes

#### 3.1.1. Preparation of the ANF and ANF Hydrogels

The preparation of ANF solution was shown in Figure 1a. In a polar solvent such as DMSO that could be saturated with a strong Lewis base, the H atoms could be removed from the N-H groups in the macromolecule chains and the whole chain could thus be converted into a polyanion, which can weaken and eliminate the interchain interactions and thus the homogenizing dispersions is achieved (Appendix A) [14]. This method was simple and effective but this process needed more than two weeks. With this in mind, we improve the dissolution rate by replacing KOH with an organic Lewis base KTB, and by introducing the proton transfer agent methanol referred to our previous work [25]. The impact of methanol was the behavior of proton transfer and the formation of methanol oligomers [27]. The size of the obtained ANF was uniform with an average diameter of 13.1 ± 2.1 nm, and the average length was more than 1.5 μm as seen in Figure 1c.

The ANF hydrogel was subsequently obtained via protonation by introducing distilled water (Figure 1a). In this process, deionized water served as the protonation reagent to precipitate ANF from solution, and the mechanism of this process is shown in Appendix A. From Figure 1d, we could see that the diameter of the ANF hydrogel in the solution was uniform with an average value of 15.8 ± 1 nm. This diameter was larger than that of the ANF fibers in DMSO, while the length was still over 1.5 μm. This behavior can be explained on the basis that with the addition of water, deprotonation occurred, and at the same time the terephthaloyl *p*-phenylenediamine polyanion picked up hydrogen to become a neutral macromolecule. Consequently, the fibers became electrically neutral. Therefore, when the fibers were in full contact with water, hydrogen bonds between macromolecule in fibers and water molecule caused the fibers to swell. Consequently, the size of the fibers was slightly increased in water.

#### 3.1.2. Preparation of the HANF Hydrogel

HANFs were obtained via the hydrolysis of the ANF hydrogel under acidic conditions [15]. In this process, carboxyl (-COOH) and amino (-NH_2_) groups were introduced, which provided these nanofibers with different hydrophilicity under different pH values. The result was confirmed by FTIR spectroscopy. The FTIR spectra of the HANFs showed similar absorption bands to those observed in the corresponding spectrum of the ANF. As shown in Figure 2b, the stretching vibration peak of N-H at 3319 cm^−1^ showed a slight shift to a lower wavenumber, which was due to the introduction of more amino groups (-NH_2_) via the hydrolysis of the amide groups (-HN-CO-). This spectral change can be attributed to the greater degree of hydrogen bonding due to the introduction of hydroxyl groups [28,29]. The absorption peaks of C=O stretching and O−H bending vibration corresponding to the -COOH were at 1652 and 1404 cm^−1^, the peak at 1546 cm^−1^ was C=O bonds of the diketone, and the peak at 1256 cm^−1^ was the stretching vibrations of C-O were visible [15]. Additionally, the SEM and AFM could prove that the HANFs still remain the morphology of fibers. From the AFM image it was evident that the diameter of the HANFs was uniform, with an average value of 11.1 ± 2.1 nm, and the length was still over 1.5 μm (Figure 1e), which was slightly smaller than the diameters of both ANF and the ANF hydrogels.

#### 3.1.3. Preparation of the ANF/HANF Membranes

As proposed in our hypothesis, the ANF/HANF membranes were obtained by vacuum-assisted filtration, and the preparation procedures are illustrated in Figure 2a. In this process, the deionized water served as the protonation reagent to precipitate ANF from solution, and then the precipitated fibers aggregated into membranes by means of the adsorption force generated by vacuum-assisted filtration. The ANF/HANF membranes obtained by this method exhibited good flexibility and high optical transparency so that the words beneath the membranes were clearly visible (Figure 2d). In addition, the thickness of these membranes was increased gradually with an increasing HANF content (Figure 2c).

### 3.2. Performance of the ANF/HANF Membranes

#### 3.2.1. pH-Responsive Characteristics of the Membranes

The pH-responsive behavior was studied by the trans-membrane fluxes and BSA retention ratios at different pH values. The water fluxes at different pH values across ANF/HANF membranes with various HANF contents are shown in Appendix A. Firstly, we could see that the flux decreased with an increasing HANF content. The reason for this trend was that the size of the HANF was smaller than that of ANF. Consequently, the HANFs could intersperse into the gaps of the ANF film as the membrane was formed and the thickness of the membrane also increased with the increase in HANF content, which was also confirmed by the BSA retention ratios and SEM images. At the pH of 7, the water flux of the pristine ANF membrane reached 151.6 L/(m^2^·h), while as the content of HANF increased to 50 wt%, the value of water flux decreased to 30.0 L/(m^2^·h), which was only 19.7% of that of the pristine ANF membranes (Appendix A). From Figure 3a,b, we could see that the trans-membrane flux and the BSA retention ratio of membranes without HANFs remained essentially unchanged in the pH range from 2.0 to 13.0, which indicated that the pure ANF films had little to no pH-responsive performance. Within the measured pH range, the trans-membrane flux initially decreased as the pH value increased, and then when the pH value exceeded 7, the water flux significant increased as the pH value continued to increase with the addition of HANFs. Meanwhile, the retention ratio of BSA initially increased while the pH value increased, and then when the pH value exceeded 7, the retention ratio of BSA decreased significantly as the pH value continued to increase with the addition of HANFs. This behavior indicated that the ANF/HANF membranes exhibited negatively pH-responsive performance in the acidic atmosphere and positively pH-responsive performance in the basic atmosphere. The change in the water flux was mainly due to the change of hydrophilia caused by the protonation and deprotonation of residual -COOH and -NH_2_. Because the intermolecular interactions were too strong, the effect of the electrostatic force caused by protonation and deprotonation of terminal -NH_2_ and -COOH groups was negligible when the content of HANF was at a low level, while the more hydrophilic -NH_3_^+^ and -COO^−^ would be produced by the -NH_2_ protonation and -COOH deprotonation. In addition, a more hydrophilic membrane will exhibit a greater water flux than its less hydrophilic counterpart [30,31,32,33,34,35]. Consequently, the water flux of films would increase under both acidic and alkaline environments. We could see that the pH-responsiveness was the most sensitive when the HANF content was 10 wt%, and when the HANF content continued to increase, the sensitivity pH-responsiveness reduced (Appendix A). The reason for this behavior might be that as the amount of -NH_2_ and -COOH increased, the electrostatic forces by protonation and deprotonation increased, which could resist the intermolecular force produced. Additionally, this reduced the pore size response of films. This effect was opposite to that of hydrophilicity on membrane water flux, so the sensitivity of the pH-responsiveness decreased with an increasing HANF content. From Figure 3a, we could see that the maximum water flux value was 113.11 L/(m^2^·h) at a pH value of 2.0 under acidic conditions. The water flux decreased as pH increased until the lowest value of 79.77 L/(m^2^·h) appeared at the pH of 7.0. As the pH continued to increase, the water flux also increased and then reached a maximum value of 120.5 L/(m^2^·h) at a pH value of 13.0. Meanwhile, the lowest BSA retention ratio achieved under acidic conditions was observed when the pH was 2.0 (94.18%), and the BSA retention ratios then became larger with an increasing pH until a maximum value was reached at a pH of 7.0 (98.88%). As the pH continued to increase, the BSA retention ratios diminished until they reached a minimum value of 93.92% at a pH of 13.0.

#### 3.2.2. Morphology of the ANF/HANF Membranes

The SEM images of the ANF and ANF/HANF membranes are presented in Figure 4, which confirmed our hypothesis. First, it was evident that membranes possessing a porous support layer were successfully prepared. From the cross-section morphology, we could see that the thickness of the ANF film was ~112 μm and that of the ANF/HANF film was ~123 μm (Figure 4c,d). Secondly, as the HANFs were added, both the surface and the interior of the film became denser. This was attributed to the fact that HANFs with smaller diameter would stay in pores formed by ANFs during the process of preparation, thus reducing the size of the pore through the membrane. Meanwhile, the surfaces of films became rougher with HANFs added, which was confirmed by the AFM measurements (Figure 5 and Table 1). The porous structure of films with the addition of the HANFs was confirmed via water flux measurements, as described in Section 3.2.3.

#### 3.2.3. Mechanical Properties

Because most of the separation membranes are carried out under pressure, the mechanical performance is a significant factor for pH-responsive membranes. Therefore, Kevlar with high mechanical strength can be used to prepare high strength pH-responsive membranes.

The image of breaking strengths and breaking elongations of ANF/HANF membranes with different HANF contents was shown in Figure 6a. The image showed that with the incorporation of HANF the tensile strength decreased. Additionally, the tensile strength decreased with the increase in the content of HANF. The tensile strength of pure ANF membrane was 112.3 MPa, and the tensile strength decreased to 55.7 MPa, when the content of incorporation HANFs was 50 wt%. The decrease could be due to the hydrolysis ofamide bonds under acidic conditions, which destroyed the original regularity and stability of the aramid fibers [29]. However, the breaking strength of the ANF/HANF membranes was still much higher than that of most reported polymer films, such as Polyethersulfone, Polysulfone, Poly(vinylidene fluoride) and Polyetheretherketones [36,37,38,39]. At the same time, the breaking elongations also decreased with the addition of HANFs. In particular, the breaking elongations decreased from 12.61% to 7.96% as the HANF contents were increased from 0 to 50 wt%. The reason for this behavior was similar to that leading to the decrease in tensile strength, which was that the HANFs were prepared via the hydrolysis of ANF, and this process resulted in weaker intermolecular forces. As a result, the breaking elongations decreased as the HANF content increased.

#### 3.2.4. Physical and Chemical Stabilities

The performance of thermal and chemical tolerance is also a significant factor to determine the practicability of filter membranes. The pH-responsive films with good thermal and chemical resistance could be used under a severe environment, and thus the range of potential applications would be greatly broadened [40]. The thermotolerance of the samples was measured via TGA test (Figure 6b). It observed that the TGA curve of ANF/HANF film was similar to that of ANF film. That was to say the addition of HANF has little influence to the thermal stability. Additionally, it observed that the initial weight reduction was due to the residual water in the membrane, and then the mass was constant with the increase in temperature before decomposition temperature. The decomposition temperatures of ANF films and ANF/HANF films were about 476 °C and 363 °C, respectively. Additionally, the initial decomposition temperatures (*T*_10_) of the pure ANF membranes and ANF/ANFs membranes were about at 495 °C and 380 °C. The reason was that the HANFs were obtained via the hydrolysis of ANF, and in this process the original regularity of the aramid fibers was destroyed to some extent, thus destroying its original crystal structure [41]. In addition, the fact that some macromolecular chains became ruptured as the amide bonds were hydrolyzed to form amino and carboxyl groups was also an important reason for the decrease in decomposition temperature [29].

The chemical stability of ANF/HANF membranes (HANF content: 10 wt%) was tested in various solvents for 15 days [42]. From Figure 7a, we could see that the mass of the membrane decreased slightly in the solvent of DMSO, DMAC, THF, DMC, while it increased slightly in the solvent of NMP, DMF, PhMe, MeOH. At the same time, the volume of the sample has a slight increase, and the rate of change was less than 0.9%. That was to say the ANF/HANF membranes offered good chemical resistance.

## 4. Conclusions

P-aramid nanofibers (ANF) with diameters of 13.1 ± 2.1 nm were successfully prepared from Kevlar via magnetic stirring in the presence of KTB and methanol for 48 h. Hydrolyzed aramid fibers (HANFs) with acid-base responsive groups (-COOH and -NH_2_) were successfully prepared via a hydrolysis reaction in the presence of H_2_SO_4_ (37.5 wt%) and HNO_3_ (8.75 wt%) by magnetic stirring for 1 h at 90 °C. Subsequently, The ANF/HANF films with dual pH responsiveness were manufactured by vacuum filtration. The ANF/HANF membranes exhibited a negative pH response under acidic atmosphere and a positive pH response under the alkaline conditions. The decomposition temperature of ANF/HANF membranes was very high, which could reach 363 °C. Interestingly, the tensile strength of ANF/HANF films was also excellent. Additionally, the tensile strength could still be maintained at 55.7 MPa even when the HANF content reached 50 wt%, which was still much larger than that of ordinary pH-responsive films made by polymer. Meanwhile, these films exhibited only a small amount of swelling after 15 days of immersion in various solvents. Therefore, such a dual-responsive membrane with excellent mechanical properties, thermal and solvent stabilities would exhibit significant potential applications such as biodegradation of wastewater, which can decrease the emission of pollution due to its responsiveness to both acidic and basic media. In addition, it also a promising candidate for other applications such as sensors, drug-controlled release, metal ion removal and protein separation.

## Figures and Tables

**Figure 1 polymers-13-00621-f001:**
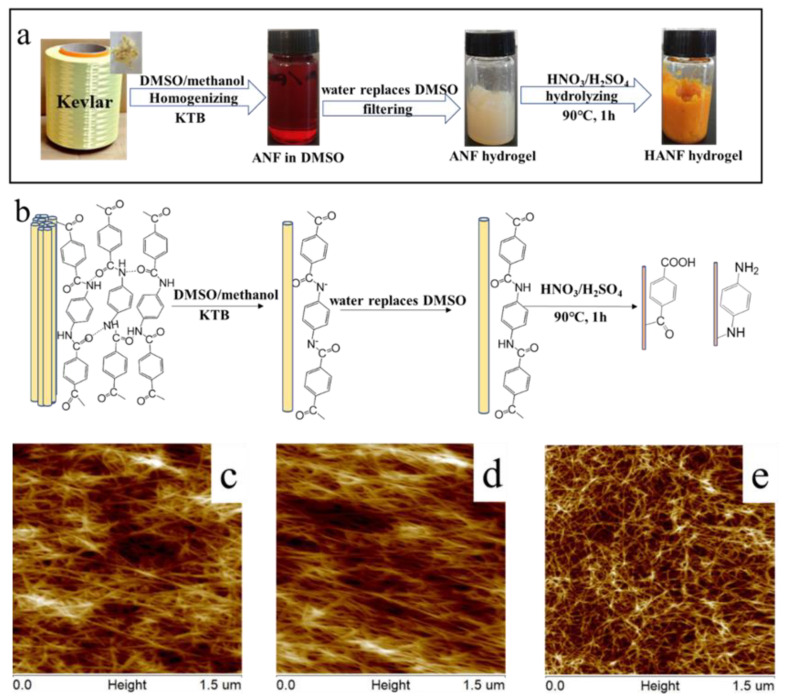
(**a**) Procedure to prepare an aramid nanofiber (ANF) solution and the hydrolyzed aramid nanofiber (HANF) hydrogel. (**b**) Synthetic mechanism leading to the ANF solution and HANF hydrogel. (**c**–**e**) AFM images of ANF in DMSO solution (**c**), ANF hydrogel (**d**), and HANF hydrogel (**e**).

**Figure 2 polymers-13-00621-f002:**
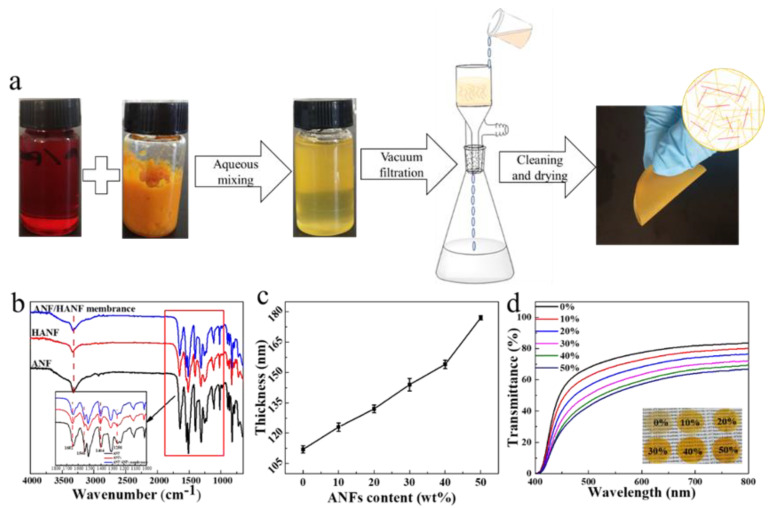
(**a**) Procedure employed to manufacture ANF/HANF membranes. (**b**) FTIR spectra of ANF, HANFs and ANF/HANF membrane (HANF composition: 10 wt%). (**c**) The change of the film thickness with the HANF content. (**d**) Transparency performance of the ANF/HANF membranes.

**Figure 3 polymers-13-00621-f003:**
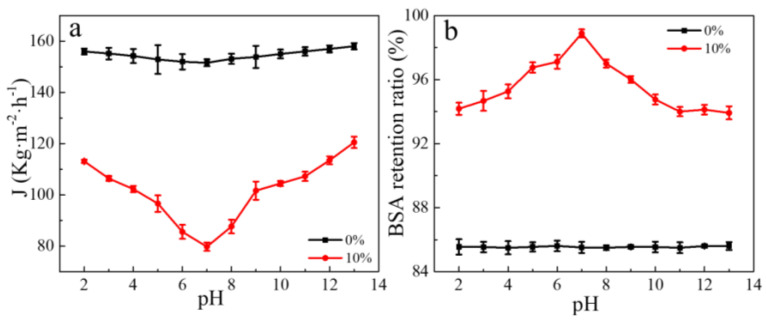
(**a**) Water fluxes at different pH values across pure ANF membranes and ANF/HANF membrane (HANF composition: 10 wt%). (**b**) BSA retention ratios at different pH values across pure ANF membranes and ANF/HANF membranes (HANF composition: 10 wt%).

**Figure 4 polymers-13-00621-f004:**
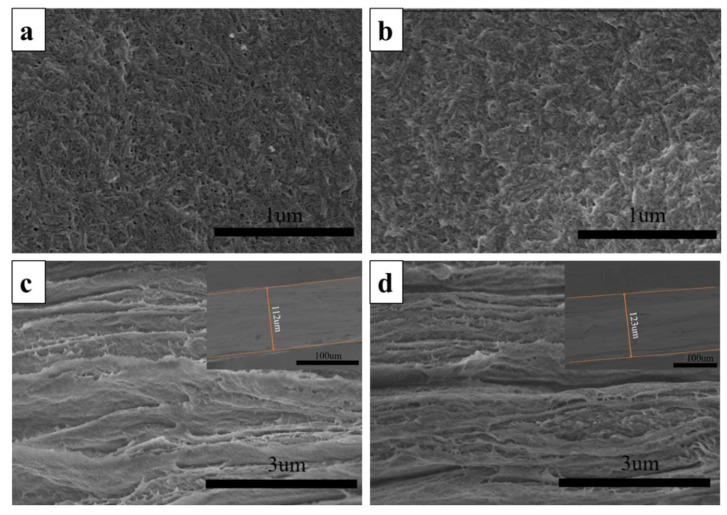
(**a**,**b**) The surface SEM images of an ANF membrane and an ANF/HANF membrance (HANF composition: 10 wt%), (**c**,**d**) SEM images of the cross-sectional of ANF membranes and ANF/HANF membranes (HANF composition: 10 wt%).

**Figure 5 polymers-13-00621-f005:**
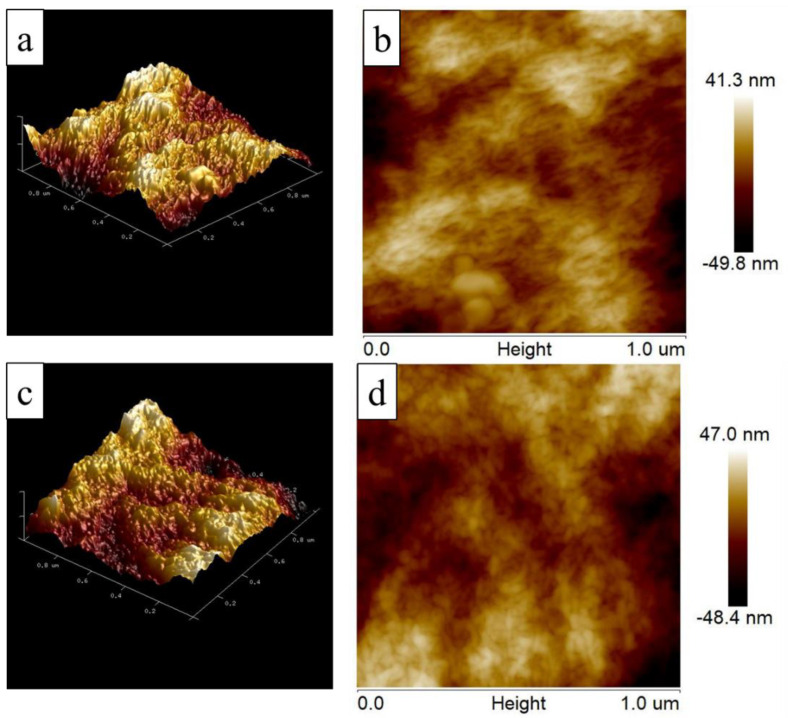
(**a**,**b**) The surface AFM images of an ANF membrane, (**c**,**d**) The surface AFM images of an ANF/HANF membrane (HANF composition: 10 wt%).

**Figure 6 polymers-13-00621-f006:**
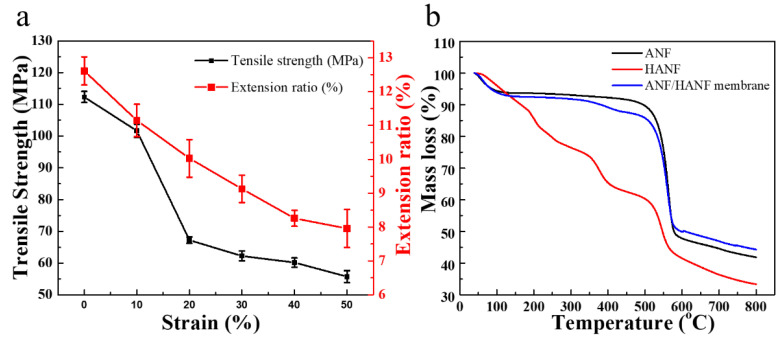
(**a**) Tensile strength and extension of membranes with various HANF contents. (**b**) The thermogravimetric analysis (TGA) curves of ANF, HANF, and ANF/HANF membrane (HANFs composition: 10 wt%).

**Figure 7 polymers-13-00621-f007:**
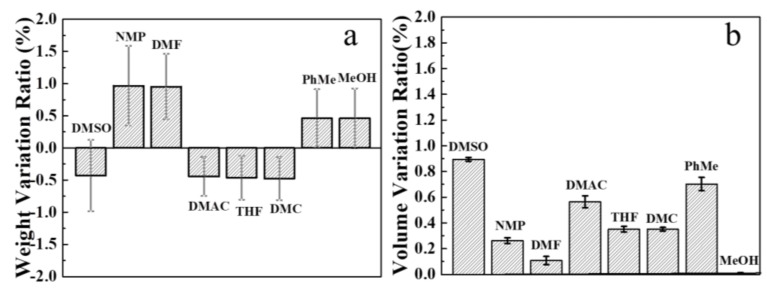
(**a**) Mass variation ratio and (**b**) swelling properties of ANF/HANF membranes (HANF content: 10 wt%) after soaking in different organic solvents for 15 days.

**Table 1 polymers-13-00621-t001:** The roughness parameters of the ANF membrane and ANF/HANF membrane (HANF composition: 10 wt%).

	*R*_q_ (nm)	*R*_a_ (nm)	*R*_max_ (nm)
ANF membrane	13.6	10.7	78.9
ANF/HANF membrane (HANFS: 10 wt%)	15.2	12.4	82.0

## Data Availability

Data available on request due to restrictions eg privacy or ethical. The data presented in this study are available on request from the corresponding author. The data are not publicly available due to this project in this area is not finished.

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
