# Peer review of "Composite Aramid Membranes with High Strength and pH-Response"

_polymers, 2021, doi:10.3390/polym13040621_

Round 1
Reviewer 1 Report
The manuscript by Wang and co-workers describe the preparation and characterization of robust aramid-based membranes. The research findings are interesting for the membrane community. The membranes were well-characterized, and a reasonable performance test was provided. The research methodology has some merits but not entirely new. However, the description of both the context of the research as well as the experimental procedures are weak and must be improved significantly. This reviewer recommends major revision to address the minor and major comments listed below.
1) The title should be simplified. What is the difference between ‘smart pH-response’ and ‘pH-response? Is there a non-smart-pH-response? Simply delete ‘smart’ from the title to avoid confusion.
2) The yield for the membrane preparation procedures under sections 2.2.1-3 should be reported. The experimental procedures should be written in more detail to aid understanding and reproduction of the work by other, e.g. instead of writing ‘plenty of water’, precisely mention the volume.
3) The state-of-the-art composite aramid membranes should be acknowledged in the introduction, the approaches compared with the reported research in the manuscript (DOI 10.1016/j.memsci.2019.117396; 10.1021/acsami.0c07341).
4) There is a recent review on the potential of aramid nanofiber composite membranes, which should be mentioned (10.1039/D0TA01654C).
5) Sections 2.3.4-5 describes the water filtration and rejection experiments. Crucial details area missing, such as the membrane area, the conditioning or pretreatment of the membranes, the duration of the filtration, the sample taking time. Were the five samples produced by independent filtrations using independently prepared membranes?
6) The purity of all solvents, chemicals, materials used in the study should be reported under the Materials sections.
7) The authors should add a short discussion on fiber-based membrane for harsh conditions that demonstrated pH-resistant and/or solvent-resistance. This topic is entirely missing from the introduction and should be covered (10.1021/acssuschemeng.9b02516; 10.1016/j.matdes.2020.108798; 10.1039/D0EN00084A; 10.1016/j.memsci.2019.01.038).
8) The authors mention the importance of mechanical performance a one of the drivers for the current research and provided reference 31 to justify it. However, this reference does not cover this topic, and mechanical performance is mentioned in it a single time only. Moreover, it does not link the mechanical properties with high operating pressures. All the membranes the authors listed that have lower mechanical properties (section 3.2.3) were successfully applied under pressure.
9) There are abbreviations that are introduced once and never used again in the entire manuscript such as ATRP, PAA; there should be deleted.
10) ‘Varition’ should be corrected to ‘Variation’
Reviewer 2 Report
Although the paper is interesting, authors should revise the paper according to folowing lines: 1- ANF, HANFs and ANF/HANF membranes were prepared in this research. Authors should conduct DSC test and compare the thermal characteristics of samples (Tg, Tm). 2- Authors have not discussed about the crystallinity and amorphous regions of samples and how there were changed during the process. XRD technique should be conducted and the crystallinity percent of samples should be calculated. 3- Authors should determine the amount of amine groups in samples by titration method.Author Response
Please see the attachment

Reviewer 3 Report
The manuscript reports the development of a new pH-responsiveness membrane by blending functional gates comprised of hydrolyzed aramid nanofibers into aramid nano-fiber membranes via a vacuum filtration method. The manuscript is within the scope of Polymers journal and is very interesting. However, before publication some concerns must be addressed namely:
- the introduction needs to be improved. Several pH-responsiveness membranes have been reported in the last few decades using different approaches. This should be discussed within the introduction.
- Section 2.2 should be improved by introducing a scheme of the main steps used for the membrane preparation. That would help the reader to get the big picture.
- Regarding the AFM data please introduce a table with the roughness parameters namely Rq, Rmax and Ra.
- The thermal analysis results may be improved. T5, Tmax and char yeld may be introduced.
- In conclusions avoid the use of words such etc. and specify the applications replacing the etc. by among others.
Round 2
Reviewer 1 Report
The authors have addressed the comments, made appropriate changes to the manuscript, which is now ready to be published in this reviewer's opinion.
Reviewer 2 Report
Authors responded my comments and revised the paper accordingly. It is acceptable now.
Reviewer 3 Report
All the issues raised were correctly and well addressed. Therefore, I support publication in the present form.